# Near-Infrared Genetically Encoded Positive Calcium Indicator Based on GAF-FP Bacterial Phytochrome

**DOI:** 10.3390/ijms20143488

**Published:** 2019-07-16

**Authors:** Oksana M. Subach, Natalia V. Barykina, Konstantin V. Anokhin, Kiryl D. Piatkevich, Fedor V. Subach

**Affiliations:** 1National Research Center “Kurchatov Institute”, Moscow 123182, Russia; 2P.K. Anokhin Institute of Normal Physiology, Moscow 125315, Russia; 3Lomonosov Moscow State University, Moscow 119991, Russia; 4MIT Media Lab, Massachusetts Institute of Technology, Cambridge, MA 02139-4307, USA

**Keywords:** calcium imaging, genetically encoded calcium indicator, protein engineering, near-infrared fluorescent bacterial phytochrome

## Abstract

A variety of genetically encoded calcium indicators are currently available for visualization of calcium dynamics in cultured cells and in vivo. Only one of them, called NIR-GECO1, exhibits fluorescence in the near-infrared region of the spectrum. NIR-GECO1 is engineered based on the near-infrared fluorescent protein mIFP derived from bacterial phytochromes. However, NIR-GECO1 has an inverted response to calcium ions and its excitation spectrum is not optimal for the commonly used 640 nm lasers. Using small near-infrared bacterial phytochrome GAF-FP and calmodulin/M13-peptide pair, we developed a near-infrared calcium indicator called GAF-CaMP2. In vitro, GAF-CaMP2 showed a positive response of 78% and high affinity (K_d_ of 466 nM) to the calcium ions. It had excitation and emission maxima at 642 and 674 nm, respectively. GAF-CaMP2 had a 2.0-fold lower brightness, 5.5-fold faster maturation and lower pH stability compared to GAF-FP in vitro. GAF-CaMP2 showed 2.9-fold higher photostability than smURFP protein. The GAF-CaMP2 fusion with sfGFP demonstrated a ratiometric response with a dynamic range of 169% when expressed in the cytosol of mammalian cells in culture. Finally, we successfully applied the ratiometric version of GAF-CaMP2 for the simultaneous visualization of calcium transients in three organelles of mammalian cells using four-color fluorescence microscopy.

## 1. Introduction

Genetically encoded calcium indicators (GECIs) engineered from green fluorescent protein-like (GFP-like) fluorescent proteins (FPs) are widely utilized for the in vivo visualization of calcium ion transients using fluorescence microscopy [1,2]. Despite the recent progress in the development of GECIs, their spectral diversity at longer wavelengths is limited by fluorescence excitation/emission maxima in the range of 561–577/589–602 nm, respectively [3].

The current trend in calcium dynamic recordings is to design GECIs that allow achieving greater imaging depths in tissue (800 μm and beyond), spectrally multiplexed imaging, and spectral compatibility with optogenetic tools. Near-infrared (NIR) fluorescence is the most attractive for the in vivo imaging because of its low light scattering, and high transparency of biological tissues in the so-called NIR transparency window in the range of 650–900 nm [4]. Calcium indicators with excitation and fluorescence emission spectra in the NIR window can be advantageous for accessing deeper tissue regions. In addition to advantages of imaging at longer wavelengths, NIR calcium indicators can also enable spectral multiplexing with numerous already existing green and red fluorescence indicators [5,6] as well as compatibility with optogenetic tools based on channelrhodopsin-2 (ChR2) [7,8].

The majority of currently available NIR fluorescent proteins exhibiting excitation/emission maxima laying in the range of 635–702/667–720 nm are derived from bacterial phytochromes [9]. Among them, the most promising are bacteriophytochrome-derived proteins which utilize biliverdin (BV) as a chromophore. BV, a product of heme oxygenation, is abundant in mammalian cells, including neurons, hence there is no need for its exogenous supplementation to enable fluorescence of phytochrome-based proteins in vivo [10]. Typically, NIR fluorescent proteins (NIR FPs) based on biliverdin-bound phytochromes consist of two domains [11]. Recently two single-domain NIR FPs, GAF-FP [12] and smURFP [13], were developed. Both of them covalently attach a BV chromophore.

To date only one NIR calcium indicator, called NIR-GECO1, utilizing bacterial phytochromes as fluorescence moiety has been reported to date by Robert Campbell’s group [14]. NIR-GECO1 exhibits an inverted response to calcium ions characterized by modest dynamic range. However, its excitation spectrum is not optimal for the commonly used 640 nm lasers. A possible reason for the limited number of NIR calcium indicators may be due to the fact that the most successful design of GECIs utilizes circularly permutated variants of GFP-like FPs fused to calcium binding domain. In the case of phytochrome-based FPs, generation of functional circularly permutated variants appears to be extremely difficult due to their N- and C-termini, unlike those in GFP-like FPs, are 40–44 Å far apart from each other as can be seen from the 3D structures of bacterial phytochromes [15,16,17]. Recently we implemented a new design of GECIs that requires insertion of calcium binding domain into FP thus eliminating the need for generation of circular permutations in the fluorescent moiety. NTnC, iYTnC, and ncpGCaMP6s indicators with the insertion of a Ca^2+^-binding domain were recently successfully developed [18,19,20]. This success prompted us to employ the same design strategy for the development of an NIR calcium indicator using bacterial phytochromes.

For the creation of an NIR calcium indicator, we decided to use GAF-FP [12] as fluorescent moiety due to its monomeric behavior (smURFP was a dimer) and minimal size of 19.6 kDa which is 2-fold and 1.4-fold smaller than other bacterial phytochrome-based NIR FPs and GFP-like proteins, respectively. As fast in vivo calcium imaging requires relatively high GECI expression levels, the small size of the novel infra-red GECI may ensure its fast maturation rate and high expression level. In addition, the small size of the NIR calcium indicator may be advantageous for the generation of fusions with individual cellular proteins or targeting to different organelles, and for delivery into cells using rAAV-particles.

Herein, we report development and characterization of a bacterial phytochrome-based NIR GECI engineered by insertion of the calmodulin/M13-peptide Ca^2+^-binding domain into the monomeric GAF-FP fluorescent protein [12]. This indicator was called GAF-CaMP2 and it was characterized by fluorescence excitation/emission maxima peaked at 642/674 nm, which made GAF-CaMP2 together with NIR-GECO1 the reddest shifted calcium indicators among all GECIs reported to date. In contrast to NIR-GECO1, the GAF-CaMP2 indicator demonstrated a positive fluorescence response to Ca^2+^ ions. When measured in vitro, its dynamic range was limited to 78% and it had affinity to calcium ions of 466 nM. GAF-CaMP2 had a 2.0-fold lower brightness and different pH stability compared to the parental GAF-FP. GAF-CaMP2 demonstrated 2.9-fold higher photostability as compared with smURFP protein. For visualization of GAF-CaMP2 in mammalian cells the addition of 20 µM BV and its fusion with sfGFP were necessary. GAF-CaMP2-sfGFP demonstrated 169% dynamic range in the cytosol of mammalian cells. Finally, we successfully applied GAF-CaMP2-sfGFP for the simultaneous visualization of [Ca^2+^] changes in three organelles of mammalian cells using four-color confocal fluorescence microscopy.

## 2. Results and Discussion

### 2.1. Development of a Novel Infra-Red Fluorescent Positive Calcium Indicator Based on GAF-FP Bacterial Phytochrome

To develop novel infrared calcium indicator we inserted the CaM/M13-peptide Ca^2+^-binding domain into five rationally selected sites of GAF-FP fluorescent bacterial phytochrome protein and performed several rounds of further optimization using directed molecular evolution in a bacterial system [21]. We selected the GAF-FP bacterial phytochrome as the fluorescent moiety for the new NIR GECI because of its small size and excitation maximum ideally matching with the 640 nm laser line available at most commercial microscopes. As a Ca^2+^-binding motif, we used the CaM and M13-peptide excised from the GCaMP6s GECI and fused by the flexible GGSSS linker in a similar way as in ncpGCaMP6s [20]. We started with the generation of five libraries with the insertion of the CaM/M13 Ca^2+^-binding part between residues 191 and 192 (L6), 204 and 205 (L7), 248 and 249 (L8), 245 and 246 (L9), or 252 and 253 (L10) of GAF-FP (Figure 1) and randomized both of 2-amino-acid-long linkers between the fluorescent GAF-FP and CaM/M13 components (Appendix A). The sites for insertion were selected according to the amino acid sequences alignment of GAF-FP and GAF domain of *Pa*BphP (Figure 1) and analysis of the available crystal structure of *Pa*BphP [17]. For GAF-FP protein we found aromatic residues such as Tyr and His which according to the *Pa*BphP X-ray structure are likely located close to BV chromophore (within a distance of 4.5–6.5 Å) and chose places for insertion close to these aromatic residues; we suggested that changing the position of the bulky aromatic residue attached to the calcium-binding domain through the linker may strongly influence the fluorescence of the BV chromophore. Screening constructed libraries on the bacterial colonies under a fluorescent stereomicroscope, we found that only library L9 had fluorescent variants. This library was further analyzed using a two-step screening strategy. During the first step, we performed imaging of the indicator’s library targeted to the *Escherichia coli* (*E. coli*) periplasm on Petri dishes and selected clones with the highest fluorescence ratio before and after treatment with a buffer that contained ethylenediaminetetraacetic acid (EDTA). During the second step, the selected clones were analyzed in bacterial extracts in B-Per reagent in a 96-well plate format. After the second step of screening, we confirmed that library L9 had clones with a positive fluorescence response to elevation of the calcium concentration showing the maximal fluorescence contrast of 1.3-fold. This variant with the highest calcium sensitivity was chosen for further optimization.

The selected clone was subjected to eight sequential rounds of random mutagenesis followed by screening. During each round, we screened approximately 20,000 colonies to identify variants with the largest Ca^2+^-dependent changes in NIR fluorescence, according to the previously described protocol [19]. After eight rounds of random mutagenesis and selection, we chose a variant with the best performance in terms of fluorescence contrast, named GAF-CaMP (GAF-FP derived CaM/M13-Peptide-based calcium indicator). The GAF-CaMP indicator had 16 mutations relative to the original template (Appendix A). We further characterized the properties of the GAF-CaMP indicator in detail in vitro (Appendix A). We did not detect fluorescence during GAF-CaMP expression in the cytosol of the HeLa cells both in the absence of BV and in the presence of 10 µM BV. However in fusion with sfGFP, in the cytosol of the HeLa cells it demonstrated positive but dramatically reduced ΔF/F dynamic range of 3.7% ± 2% (mean ± SD throughout the paper) upon 2.5 µM ionomycin addition by 14- and 26-fold as compared with its ΔF/F response of 52% ± 15% and 95% ± 9% for the purified protein and bacteria, respectively (Appendix A Results and Discussion). We suggested that because of too high affinity to calcium ions (K_d_ = 15 ± 2 nM), GAF-CaMP was saturated in the cytosol of mammalian cells at 50–100 nM physiological calcium concentrations [22,23] and does not seem to be suitable for the commonly used types of the mammalian cells and neuronal imaging. However, some cells such as amoeba *Dictyostelium discoideum* have low resting calcium concentrations and indicator YC-Nano that had K_d_ 15 nM identical to that for GAF-CaMP was successfully applied to visualize calcium activity in these cells in vivo [24]. Amoeba and amoeba-like cells have quite interesting forms of behavior and cellular plasticity studies of which might benefit from applying calcium imaging [25,26,27]. Hence, GAF-CaMP may be appropriate for the detection of calcium transients in the cells with low resting calcium concentrations.

The inability of GAF-CaMP indicator to sense calcium transients in the commonly used mammalian cells prompted us to subject GAF-CaMP to further mutagenesis with the aim of reducing its affinity to calcium ions and increasing its dynamic range in mammalian cells. As the linker between CaM and M13-peptide affected the calcium affinity and kinetics for Cameleon-Nano indicators [24], we first increased the length of the linker between CaM and M13-peptide by insertion of GGGS-peptide (Figure 2 and Figure 3a).

The generated GAF-CaMP derivative was further subjected to nine sequential rounds of random mutagenesis followed by screening as described above with some modifications. To facilitate colonies screening on Petri dishes we expressed GAF-CaMP in bacteria in fusion with sfGFP. To select mutants with decreased calcium affinity we shortened the time after EDTA spraying on Petri dishes from 24 till 1 h and selected clones with increased K_d_ according to the characterization of protein on bacterial lysates. After nine rounds of random mutagenesis and selection, we chose a variant with the best performance in terms of fluorescence contrast and calcium affinity, named GAF-CaMP2.

The GAF-CaMP2 indicator had 41 mutations relative to the original template (Figure 2) or 25 mutations vs. GAF-CaMP indicator. Among these mutations, 22, 15, and four mutations were located in the fluorescent part, CaM/M13-peptide calcium-binding part, and linkers between these parts, respectively. According to the amino acid sequence alignment of GAF-FP and *Pa*BphP (Figure 1) and crystal structure of *Pa*BphP [17], the S106P, M107T, T302S, and H320Y mutations in the fluorescent domain were located 4.5–5.5 Å close to the BV chromophore and 18 other mutations were more than 6.5 Å away from BV. We suggested that S106P, M107T, T302S, and H320Y mutations in the fluorescent GAF-domain might be responsible for the brightness, dynamic range, and affinity to BV of GAF-CaMP2 indicator through the direct interaction between polypeptide and BV chromophore. The D239A or I137V, I173L, and F175Y mutations in the CaM domain were positioned at calcium-binding residue or next to the calcium-binding residues, respectively, and might affect the affinity of GAF-CaMP2 indicator to calcium ions. Mutations in M13-peptide such as K270M and T274A might also affect the affinity of GAF-CaMP2 indicator to calcium ions in a similar way as for the FGCaMP indicator [28].

### 2.2. In Vitro Characterization of the Purified GAF-CaMP2 Indicator

First, we characterized the spectral and biochemical properties of the purified GAF-CaMP2 calcium indicator alone in the Ca^2+^-saturated (sat) and Ca^2+^-free (apo) states (Figure 3 and Table 1). In parallel, we characterized GAF-CaMP2-sfGFP fusion (Figure 4 and Table 1) because it was fluorescent in mammalian cells, as described below.

At pH 7.2, GAF-CaMP2_sat_ and GAF-CaMP2_apo_ had the most red-shifted absorption peak at 649–650 and 630–642 nm, respectively (Figure 3b and Figure 4b). The Soret band was observed for both states at 383–384 nm. The excitation maxima in sat- and apo-states were at 640–642 and 620–630 nm, respectively (Figure 3c and Figure 4c). When excited at 640 nm, GAF-CaMP2_sat_ and GAF-CaMP2_apo_ fluoresced with an emission peak at 674 and 676 nm, respectively (Figure 3c and Figure 4c). The brightness of the GAF-CaMP2_sat_ indicator in terms of the product of the extinction coefficient and quantum yield was 2.0–2.6-fold lower than that of the GAF-FP progenitor (Table 1). In the absence of calcium ions, the brightness of GAF-CaMP2_apo_ dropped as a result of a decrease of both quantum yield and extinction coefficient. The fluorescence dynamic range of the GAF-CaMP2 indicator was 93%–103% (Table 1). Addition of 1 mM physiological concentration of Mg^2+^ ions decreased the dynamic range of GAF-CaMP2 up to 77%–78%, which was significantly lower than that for the commonly used indicators such as red R-GECO1 (16-fold intensity change) [31] and inverted NIR-GECO1 (8-fold contrast) [14].

As the physiological pH in the cells may vary from 5.0 in lysosomes till 7.5 in the cytoplasm [32], we assessed the dependence of fluorescence and dynamic range of the GAF-CaMP2 indicator on pH. In the presence of 5 mM calcium ions, GAF-CaMP2_sat_ exhibited a pH-dependence of its fluorescence with p*K*_a_ values 4.89, 7.15–7.60 and ≥9.1−9.31 which were different from those for GAF-FP progenitor and smURFP proteins (Figure 3d, Figure 4d, Appendix A and Table 1). The fluorescence of GAF-CaMP2_apo_ changed to variations of pH with p*K*_a1_ = 5.16–5.3, p*K*_a2_ = 6.99–7.28 and p*K*_a3_ ≥ 9.28–9.30. The different pH-stability of GAF-CaMP2 in Ca^2+^-bound and Ca^2+^-free states resulted in dependence of its dynamic range from pH. The NIR-GECO1 indicator revealed higher pH-sensitivity than GAF-CaMP2 with p*K*_a_ values of 6 and 9 [14]. The commonly used red R-GECO1 (p*K*_a_ = 6.59 [31]) and green GCaMP6s (p*K*_a_ = 6.16 [18]) calcium indicators based on the GFP-like proteins also revealed the sensitivity of their fluorescence to pH variations within the pH range of 5–7.5. Thus, pH variations may contribute to the GAF-CaMP2 Ca^2+^ response in a similar way as for the calcium indicators based on GFP-like proteins.

We further determined the affinity of the GAF-CaMP2 indicator to Ca^2+^ ions in the absence and in the presence of 1 mM Mg^2+^, a concentration that resembles that in the cytoplasm of mammalian cells [33]. It is known that the free calcium concentration may vary in the range of 50–100 nM to 250–10,000 nM in the cytoplasm of mammalian cells [22,23]. According to the equilibrium binding titration experiments, GAF-CaMP2 demonstrated a K_d_ value of 235–289 nM (Figure 3e, Figure 4e, and Table 1). Addition of 1 mM Mg^2+^ ions increased K_d_ value up to 435–466 nM or in 1.6–1.9-fold, which was 1.4-fold less than the respective constant for the GCaMP6f indicator in the presence of 1 mM Mg^2+^ ions (K_d_ = 632 nM); GCaMP6f demonstrated similar 1.7-fold decrease of its calcium affinity in response to the 1 mM Mg^2+^ addition. The equilibrium Hill coefficient for the GAF-CaMP2 indicator (*n* = 0.88–1.54) was lowered as compared to the GCaMP6f (*n* = 2.25–2.4) or GCaMP6s GECI (*n* = 4.0 [18]) in a similar way as for ncpGCaMP6s [20], which provides evidence for the decreased cooperativity of Ca^2+^ binding by GAF-CaMP2. The difference in Hill coefficient values between GAF-CaMP2 and GCaPM6f/s could be attributed to the direct linkage of CaM and M13-peptide, which affects the cooperativity of their Ca^2+^-dependent interaction.

The affinity of GAF-CaMP2 indicator to BV chromophore was further characterized for both apo- and sat-states. The GAF-CaMP2-sfGFP protein purified in the absence of BV chromophore did not reveal absorbance around 383 and 650 nm suggesting that BV chromophore was washed out during the purification procedure. Titration of GAF-CaMP2-sfGFP protein with BV in the presence of 5 mM Ca^2+^ ions resulted in the appearance of far-red fluorescence with apparent K_d_ value of 28 ± 10 nM (Figure 4f). The affinity of GAF-CaMP2 indicator to BV in the absence of calcium decreased to a K_d_ value of 169 ± 52 nM or 6-fold. In both states, GAF-CaMP2 had BV affinity higher than those for iRFP (350 nM) and IFP1.4 (4200 nM) bacteriophytochrome-based permanently fluorescent proteins [34]. Hence, the affinity of GAF-CaMP2 indicator to BV chromophore was 6-fold different for the sat- and apo-states but for both states it was in the nM-range and 2–25-fold higher for the apo-state than those for iRFP and IFP1.4 proteins.

We also characterized the photostability of GAF-CaMP2 using smURFP protein with similar spectral characteristics as a control. Under a wide-field near-infrared light illumination (Ex: 620/60 BP, Em: 700/75 BP, the power of light before 63 × 1.4 NA oil immersion objective lens: 2.23 mW/cm^2^), the GAF-CaMP2 photobleached 2.9–3.1-fold slower than smURFP protein (Figure 3f, Figure 4g, and Table 1). Hence, the high photostability of GAF-CaMP2 indicator can to some extent compensate its low brightness during imaging.

We finally assessed the maturation rate of GAF-CaMP2 using sfGFP as an internal control. At 37 °C, the GAF-CaMP2 indicator in the Ca^2+^-saturated state had 1.6-fold slower maturation rate than sfGFP (Figure 3g and Table 1). Maturation rate of GAF-CaMP2 was practically similar to or 5.5-fold faster than those for smURFP [13] and GAF-FP [12] proteins, respectively.

Overall, the in vitro characterization indicated that GAF-CaMP2 had 5–7/4–6 nm red-shifted excitation/emission maxima and 2.0–2.6-fold lower brightness as compared with GAF-FP progenitor protein; as compared with inverted NIR-GECO1 indicator, GAF-CaMP2 showed higher pH stability and demonstrated a 9-fold lower but positive ΔF/F response of 77%–78% to calcium ions with affinity of 435–466 nM; it revealed a 2.9–3.1-fold higher photostability than smURFP protein. The main advantage of the GAF-CaMP2 indicator is that it has an excitation maximum at 640–642 nm which is optimal for the excitation with the standard 633–640-nm red lasers used in flow cytometers and confocal microscopes.

### 2.3. Optimization of Expression and Calcium-Dependent Response of the GAF-CaMP2 Indicator in HeLa Mammalian Cells

To characterize the behavior of the GAF-CaMP2 indicator in mammalian cells, we investigated its expression and response to the Ca^2+^ transients in HeLa Kyoto cells. The NES-GAF-CaMP2 indicator transiently co-expressed with sfGFP from different plasmids in HeLa cells was practically non-fluorescent even in the presence of 20 µM BV (Figure 5a). To address this issue, we generated the NES-GAF-CaMP2-sfGFP fusion (Appendix A) and transiently expressed it in HeLa cells. In the presence of 20 µM BV supplied during transfection procedure 24–48 h before imaging, the GAF-CaMP2-sfGFP fusion revealed far-red fluorescence evenly distributed in the cytosol of the cells (Figure 5b). Addition of 2.5 µM ionomycin to the cell culture resulted in an increase of GAF-CaMP2 fluorescence with averaged ΔF/F response of 169% ± 56%; the green fluorescence signal of sfGFP was not affected by the ionomycin addition (Figure 5b and Table 1). The ΔF/F response of GAF-CaMP2 in the cytosol of the cells was 2.2-fold larger as compared with its dynamic range for the purified protein (Table 1). In the absence of BV, NES-GAF-CaMP2-sfGFP fusion was non-fluorescent in HeLa cells both at low and high concentrations of calcium ions (Appendix A). Hence, the expression of GAF-CaMP2 indicator in mammalian cells demands its C-terminal fusion with sfGFP and 20 µM BV chromophore supply and under these conditions, GAF-CaMP2 demonstrated positive ΔF/F response 169% ± 56% to the ionomycin-induced calcium concentration elevation.

### 2.4. Simultaneous Visualization of Calcium Transients in Three Organelles of Mammalian Cells using GAF-CaMP2 Indicator and Four-Color Fluorescence Confocal Microscopy

GAF-CaMP2 had excitation/emission maxima spectrally distinct from commonly used blue B-GECO1 and red R-GECO1 GFP-based GECIs. Therefore, this prompted us to demonstrate simultaneous visualization of calcium transients in three organelles of HeLa cells using these indicators and multi-color fluorescence confocal microscopy. With this aim, we transiently co-expressed blue H2B-B-GECO1, red IMS-R-GECO1, and green/far-red GAF-CaMP2-sfGFP indicators in the nucleus, intermembrane space of mitochondria, and cytosol of the HeLa cells. A total of 20 µM BV was supplied during the transfection procedure. The addition of 2.5 µM ionomycin induced a virtually simultaneous increase of blue nuclei-localized and far-red cytosolic-localized fluorescence of B-GECO1 and GAF-CaMP2 indicators, respectively, but the IMS-localized red fluorescence increase for R-GECO1 was slightly delayed (Figure 6a and Appendix A).

We also attempted to simultaneously visualize the calcium transients in these organelles induced by 10 μM thapsigargin, which evokes Ca^2+^ ion release from intracellular Ca^2+^ stores, such as the ER and its sub-compartments, via inhibition of Ca^2+^-ATPases in the sarco-endoplasmic reticulum [35]. The addition of 10 µM thapsigargin resulted in an increase of fluorescence in all organelles but with slightly delayed red fluorescence dynamics in case of IMS (Figure 6b and Appendix A). The delayed calcium dynamics for IMS-R-GECO1 indicator in both ionomycin- and thapsigargin-induced calcium transients may be attributed to its lower affinity or sensitivity to calcium ions (K_d_ = 482 nM) as compared with those for B-GECO1 (K_d_ = 164 nM) and GAF-CaMP2 (K_d_ = 289 nM). Thus, proper calcium affinity and spectrally distinct excitation/emission maxima of GAF-CaMP2 allow its combined utilization with blue and red GFP-based GECIs for simultaneous visualization of calcium transients in three different organelles of mammalian cells.

## 3. Materials and Methods

### 3.1. Mutagenesis and Library Screening

The GAF-FP protein was synthesized using polymerase chain reaction (PCR) with overlapping primers (Appendix A) as described previously [36]. Library construction and screening were performed as described in [19] with some modifications. Libraries were expressed in the *E. coli* BW25113 strain with the pWA23 plasmid encoding heme oxygenase-1 under the control of a rhamnose-inducible promoter to enable BV production. Near-infrared fluorescence of colonies during a primary screening of bacterial libraries was registered with ET 620/60 X excitation at 67 µW/cm^2^ light density and ET 700/75 M emission filters (Chroma, Bellows Falls, VT, USA), respectively. Expression of the indicators and heme oxygenase-1 in the colonies on Petri dishes was induced with 0.0002% arabinose and 0.002% rhamnose, respectively. Colonies were grown at 30 °C for 48 h. Screening using a 96-well plate reader was performed on bacterial protein extracts. For lysates production bacteria expressing GAF-CaMP variants were pelleted down from 10 mL of LB medium supplemented with ampicillin (100 µg/mL), kanamycin (30 µg/mL), 0.004% arabinose, and 0.04% rhamnose for 24 h at 37 °C, 24 h at r.t. and 220 rpm. Proteins were further extracted with 300 µL of B-Per extraction reagent (Thermo Scientific, Rockford, IL, USA) supplemented with lysozyme (1 mg/mL) and DNase I (4 U/mL, Invitrogen, Waltham, MA, USA) by incubation at 37 °C at 220 rpm followed by vortexing for 10 s and centrifugation at 14,800 rpm for 1 min.

### 3.2. Protein Purification and Characterization

Proteins were expressed in BW25113/pWA23 bacteria grown in LB medium supplemented with ampicillin (100 µg/mL), kanamycin (30 µg/mL), 0.004% arabinose, and 0.04% rhamnose for 6 h at 37 °C, 12 h at 30 °C and 24–48 h at 18 °C and 220 rpm. Protein purification was further performed as described in [19] except proteins were eluted from the Ni-NTA resin with 100 mM EDTA, pH 7.4, 20 mM Tris-HCl. In the case of GAF-CaMP2 and GAF-CaMP2-sfGFP fusion proteins 1 µM BV was supplied during all stages of purification and characterization.

The extinction coefficient value for the purified GAF-CaMP protein in the Ca^2+^-saturated state was calculated in buffer 30 mM HEPES, pH 7.2, 100 mM KCl (buffer A) supplemented with 10 mM CaCl_2_, relative to the peak at 380 nm that had an extinction coefficient of 39,900 M^−1^·cm^−1^ [12]. The extinction coefficients for the GAF-CaMP2 and GAF-CaMP2-sfGFP fusion proteins were determined in the same buffer A supplemented with 5 mM CaCl_2_ and 1 µM BV, relative to the peak at 380 nm that had an extinction coefficient of 39,900 M^−1^·cm^−1^ [12] for GAF-CaMP2 or relative to the peak at 490 nm that had an extinction coefficient of 56,000 M^−1^·cm^−1^ (characteristic for GFP form) for GAF-CaMP2-sfGFP. Alternatively, extinction coefficients were estimated relative to the peak at 280 nm; the extinction coefficients for GAF-CaMP2 (33,725 M^−1^·cm^−1^) and GAF-CaMP2-sfGFP (86,610 M^−1^·cm^−1^) proteins at 280 nm were calculated using the ProtParam tool online service.

For quantum yield determination, the integrated fluorescence values (in the range of 624–773 nm) of purified GAF-CaMP in the Ca^2+^-saturated state excited at 590 nm were measured in buffer A supplemented with 10 mM CaCl_2_ and compared with the same values for the equally absorbing at 590 nm GAF-FP protein. The quantum yields of GAF-CaMP2 and GAF-CaMP2-sfGFP fusion proteins were determined in the same buffer A supplemented with 5 mM CaCl_2_ and 1 µM BV for Ca^2+^-saturated form and with 10 mM EDTA and 1 µM BV for apo-form. For forms with excitation maxima at 620–642 nm the integrated fluorescence values of purified GAF-CaMP2 (in the range of 620–820 nm) and GAF-CaMP2-sfGFP (in the range of 600–820 nm) excited at 590 nm were measured and compared with the same values for the equally absorbing at 590 nm miRFP720 [5]. For control smURFP protein the integrated fluorescence values of purified smURFP (in the range of 620–820 nm) excited at 590 nm were measured and compared with the same values for the equally absorbing at 590 nm miRFP720 [5]. In all cases for GAF-CaMP2 and GAF-CaMP2-sfGFP integrated fluorescence values of buffers supplemented with 1 µM BV in the same ranges were subtracted from integrated fluorescence values of protein solutions.

For equilibrium calcium K_d_ determination, the fluorescence of the GAF-CaMP protein (2 ug/mL final concentration) was measured in the mixture of two stock buffers A containing 10 mM EGTA or 10 mM Ca-EGTA, as described previously [37]. In case of the GAF-CaMP2 and GAF-CaMP2-sfGFP fusion proteins (50 nM final concentration) for K_d_ determination, we used the same buffers, except 30 mM HEPES was replaced with 30 mM MOPS and 1 µM BV was additionally added.

For equilibrium BV K_d_ determination, the GAF-CaMP2-sfGFP protein was purified as described above but in the absence of BV on all stages. The far-red fluorescence of GAF-CaMP2-sfGFP protein (50 nM final concentration) was further measured in buffer A supplemented with 5 mM CaCl_2_ (sat-state) or 10 mM EDTA (apo-state) and varying concentrations of BV: 0, 10, 20, 40, 80, 100, 200, 400, 800, 1000, and 2000 nM.

To measure dissociation kinetics, protein solution in buffer A was rapidly mixed (1:100) with buffer A supplemented with 0.1 mM EGTA. Fluorescence was registered with a CM2203 spectrofluorometer (Solar, Minsk, Belarus).

pH titrations were performed using incubation of purified proteins in buffers 30 mM citric acid, 30 mM borax, 30 mM NaCl with pH ranging from 3.0 to 10.0 for 20 min at r.t. as described in [18]. The pH stability of GAF-CaMP2 and GAF-CaMP2-sfGFP fusion proteins (50 nM final concentration) was determined in the same buffers supplemented with 1 µM BV.

Maturation of GAF-CaMP2-sfGFP fusion protein was performed according to the previously published [37] procedure with some modifications. In brief, BW25113/pWA23 bacteria transformed with the pBAD/HisB-GAF-CaMP2-sfGFP plasmid were grown in 250 mL of LB medium supplemented with ampicillin (100 µg/mL), kanamycin (30 µg/mL), and 0.02% rhamnose at 37 °C overnight. The next morning, 0.1% arabinose was added to bacterial cells. Upon induction of protein expression, bacterial cultures were grown at 37 °C in 50 mL tubes filled to the brim and tightly sealed to restrict oxygen supply. After 2 h, the cultures were centrifuged and sonicated in PBS buffer supplemented with 5 mM imidazole and 1 µM BV and the resulting protein was purified using Ni-NTA resin within ~20 min, with all procedures and buffers at or below 4 °C in the presence of 1 µM BV. Protein was eluted from Ni-NTA resin with 100 mM EDTA, 20 mM Tris-HCl, pH 7.40, 1 µM BV to avoid bound BV chromophore displacement from the protein by high concentrations of imidazole. Protein maturation occurred in buffer A supplemented with 5 mM Ca-EDTA and 1 µM BV, at 37 °C. Green (ex./em. 460/520 nm, respectively) and far-red (ex./em. 640/675 nm, respectively) fluorescence signals of the protein were monitored using a CM2203 spectrofluorometer (Solar, Minsk, Belarus).

Photobleaching experiments were performed on the suspension of purified proteins (4 µM concentration in buffer A supplemented with 10 mM CaCl_2_ (GAF-CaMP1) or 5 mM CaCl_2_, 1 µM BV (GAF-CaMP2 and GAF-CaMP2-sfGFP) in mineral oil, as previously described [37], with some modifications. Briefly, photobleaching was performed using an inverted Nikon Eclipse Ti-E/B microscope (Nikon Instruments, Tokyo, Japan) equipped with a laser spinning-disk Andor WD Technology Revolution multi-point confocal system (Andor Technology Ltd., Belfast, UK), a 75 W mercury-xenon lamp (Hamamatsu Photonics, Hamamatsu, Japan), a 60× oil immersion objective NA 1.4 (Nikon, Tokyo, Japan), a 16-bit Neo sCMOS camera (Andor Technology Ltd., Belfast, UK), laser module Revolution 600 (Andor Technology Ltd., Belfast, UK), spinning-disk module Yokogawa CSU-W1 (Andor Technology Ltd., Belfast, UK). The near-infrared fluorescence was bleached and acquired using 620/60 excitation light from a mercury-xenon lamp (2.23 mW/cm^2^ power before objective lens), confocal dichroic mirror 660 and emission filter 700/75 (Nikon Instruments, Tokyo, Japan). No corrections were performed for calculation of photostability.

In vitro stabilities of purified proteins (50 nM final concentration) were measured in 30 mM HEPES, pH 7.40 buffer supplemented with 1 µM BV and varying concentrations of guanidine hydrochloride (0.5, 1.0, 1.5, 2.0, 2.5, 3.0, 3.5, 4.0, 4.5, and 6.0 M). Far-red and green fluorescence were registered using 96-well ModulusTM II Microplate Reader (Turner Biosystems, Sunnyvale, CA, USA) after incubation at 25 °C during 20 h.

### 3.3. Mammalian Live-Cell Imaging

HeLa Kyoto cell cultures were imaged 24−48 h after transfection in the presence of varying concentrations of BV using a laser spinning-disk Andor WD Technology Revolution multi-point confocal system (Andor Technology Ltd., Belfast, UK) as previously described [37].

For time-lapse imaging experiments with varying Ca^2+^ concentrations, 2.5 μM ionomycin or 10 μM thapsigargin were added to the cells during imaging.

### 3.4. Mammalian Plasmid Construction

In order to construct the pAAV-*CAG*-NES-GAF-CaMP(2) and pAAV-*CAG*-NES-GAF-CaMP(2)-sfGFP plasmids, the GAF-CaMP(2) and GAF-CaMP(2)-sfGFP genes were PCR amplified as the BglII-EcoRI and BglII-HindIII fragments using primers listed in the table (Appendix A), respectively, and swapped with the mCherry gene in the pAAV-*CAG*-NES-mCherry vector. In order to construct pAAV-*CAG*-H2B-B-GECO1 and pAAV-*CAG*-IMS-R-GECO1 plasmids, B-GECO1 and R-GECO1 genes were PCR amplified as the BglII-HindIII and BglII-EcoRI fragments and swapped with the mCherry and YTnC genes in the pAAV-*CAG*-H2B-mCherry and pAAV-*CAG*-IMS-YTnC vectors, respectively.

## 4. Conclusions

In conclusion, we developed a novel genetically encoded near-infrared GAF-CaMP2 calcium indicator based on the minimal domain of bacterial phytochrome and demonstrated its applicability for live-cell multicolor confocal imaging of calcium transients in different cellular organelles. Though GAF-CaMP2 (351 a.a.) has 1.4-fold smaller molecular size than NIR-GECO1 (496 a.a.), its expression in mammalian cells demands fusion with sfGFP making it 1.2-fold larger in size (593 a.a.). Owing to spectrally distinct excitation/emission maxima of GAF-CaMP2 (ex./em. 642/674 nm) and NIR-GECO1 (ex./em. 678/704 nm), GAF-CaMP2 could be used in combination with NIR-GECO1 and provide additional color.

GAF-CaMP2 and NIR-GECO1 have positive and inverted responses to calcium ions, respectively. To suggest possible structural bases for such an opposite response in two sensors, we aligned the X-ray structures of the *Pa*BphP and *Dr*BphP bacteriophytochromes and found that the insertion sites of calcium-binding domains for GAF-CaMP2 and NIR-GECO1 indicators are located on opposite sides of the BV chromophore (ESI, Appendix A). The opposite arrangement of calcium-binding domains relative to the BV chromophore for GAF-CaMP2 and NIR-GECO1 may be one of the possible explanations for the opposite calcium response for these indicators.

The fusion of the GAF-CaMP2 indicator with sfGFP was fluorescent in mammalian cells, but the GAF-CaMP2 alone was not. We suggest several possible reasons for the extremely weak fluorescence of GAF-CaMP2 indicator in mammalian cells related to its possible poor folding, low stability, or susceptibility to degradation by cellular machinery. It is known that sfGFP protein or other well-folded proteins may facilitate folding of proteins in fusion with them [29]. In accordance with this assumption, the extinction coefficient for bacterially expressed GAF-CaMP2 protein (3987 M^−1^ cm^−1^ at 650 nm) calculated relative to the absorption peak at 280 nm was 2.6-fold smaller as compared to the extinction coefficient for GAF-CaMP2-sfGFP fusion (10,192 M^−1^ cm^−1^ at 649 nm). Hence, in bacterial cells GAF-CaMP2 protein folds 2.6-fold less efficiently as compared to its fusion with sfGFP. Therefore, GAF-CaMP2 alone may fold inefficiently in mammalian cells in similar way as in bacteria and sfGFP fused to its C-terminal end may improve its folding by 2.6 times. High cellular stability of bacteriophytochrome-based proteins can also correlate with their high protein stability in vitro [38]. Respectively, the possible poor stability of GAF-CaMP2 protein may also affect its brightness in mammalian cells. We compared the in vitro stability of GAF-CaMP2, GAF-CaMP2-sfGFP, and smURFP proteins according to resistance of their fluorescence to guanidine hydrochloride denaturating reagent. We found that both GAF-CaMP2 and GAF-CaMP2-sfGFP had higher in vitro stability as compared to smURFP protein (Appendix A). GAF-CaMP2 demonstrated in vitro stability in fusion with sfGFP, similar to its stability alone (Appendix A). Hence, GAF-CaMP2 itself has high stability in vitro and sfGFP protein practically does not affect it and cannot increase expression level of fused GAF-CaMP2 protein in mammalian cells by increasing of GAF-CaMP2 stability. Some peptides, called degrons, in fusion with protein of interest (POI) can modulate the intracellular POI concentration through the susceptibility to degradation via cellular proteasome system [39]. sfGFP may serve as a stabilizing degron tag and elongate half-life of the GAF-CaMP2 protein in mammalian cells. Hence, sfGFP enhances expression level of fluorescent GAF-CaMP2 indicator in mammalian cells, probably, by improving folding of GAF-CaMP2 and/or the decreasing its susceptibility to degradation.

We anticipate that further enhancement of the GAF-CaMP2 indicator brightness, its affinity to BV chromophore, and protein folding/degradation in mammalian cells should bring it to the next performance level. We believe that the molecular design suggested in this paper for the development of a near-infrared GAF-CaMP2 calcium indicator will help and prompt other studies on the generation of other near-infrared calcium indicators based on bacterial phytochromes.

## Figures and Tables

**Figure 1 ijms-20-03488-f001:**
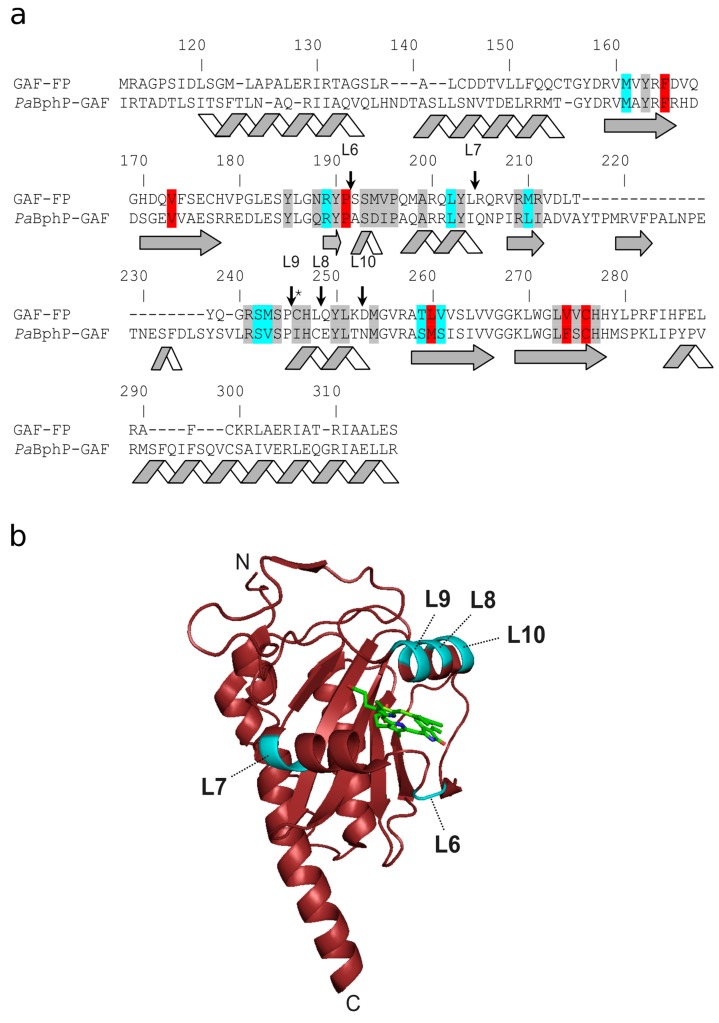
Alignment of the amino acid sequences of GAF-FP and GAF domain of *Pa*BphP (*Pa*BphP-GAF) and representation of insertion sites of calcium binding domain. (**a**) Alignment numbering follows that of PaBphP. The residues which are within 4.5, 4.5−5.5, and 5.5−6.5 Å surrounding the biliverdin (BV) chromophore according to the X-ray structure of PaBphP (3C2W) are highlighted in grey, cyan, and red colors, respectively. Stars indicate Cys-residue that is covalently bound to the chromophore. Sites of insertion in the GAF-FP protein are indicated with arrows. (**b**) Insertion sites of calcium-binding domain are highlighted in cyan on X-ray structure of GAF domain (PDB 3C2W). The BV chromophore is shown in green.

**Figure 2 ijms-20-03488-f002:**
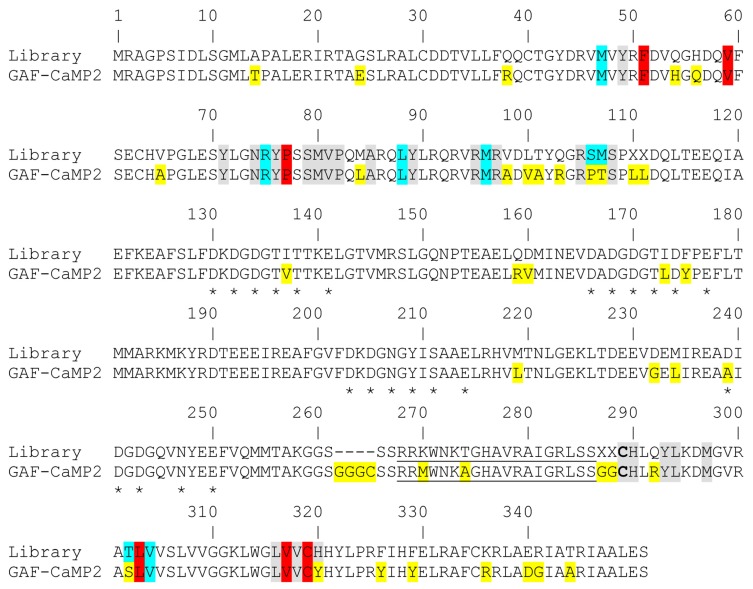
Alignment of the amino acid sequences for the original library L9 and GAF-CaMP2 calcium indicator. Alignment numbering follows that of original library L9. Mutations in GAF-CaMP2 related to the initial library L9 including linkers between fluorescent and indicator parts are highlighted in yellow. The residues which are suggested within 4.5, 4.5–5.5, and 5.5–6.5 Å surrounding the BV chromophore according to the X-ray structure of PaBphP (3C2W) are highlighted in grey, cyan, and red colors, respectively. Residues in the CaM-part that assumed to bind Ca^2+^ ions are selected with stars. M13-peptide is underlined. Cys-residue 289 that is suggested to be covalently bound to the chromophore of GAF-FP protein is selected with bold.

**Figure 3 ijms-20-03488-f003:**
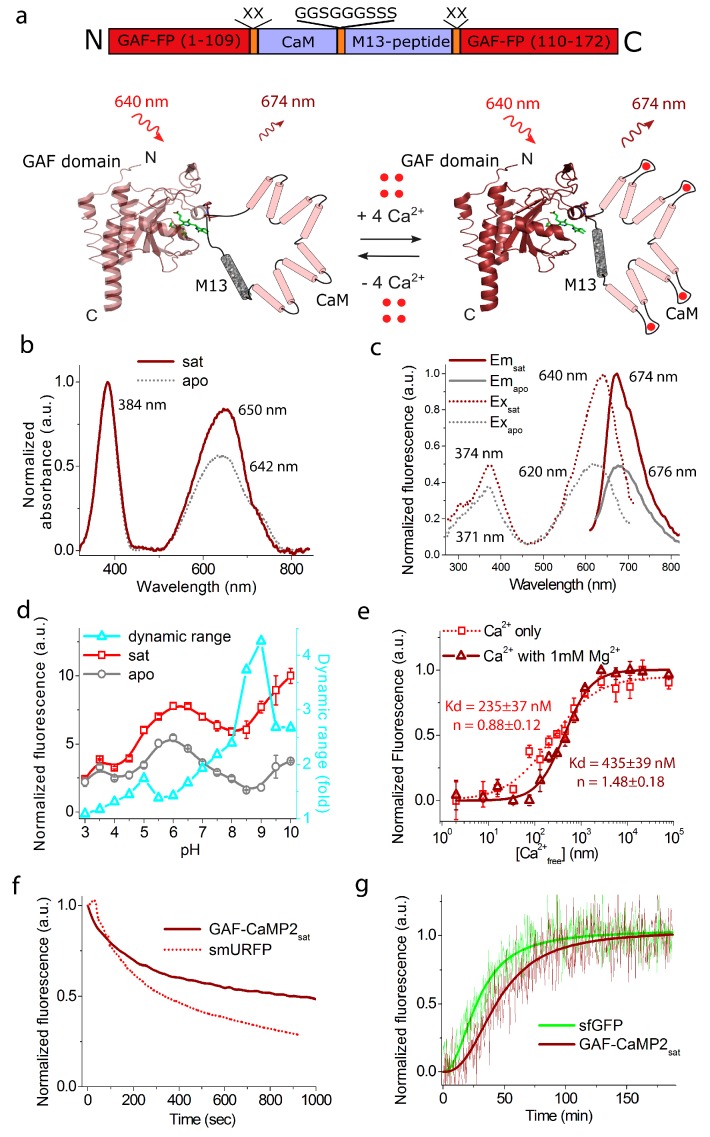
In vitro properties of the purified GAF-CaMP2 indicator. (**a**) A scheme of the original library for optimization of linkers in the GAF-CaMP2 indicator and suggested mechanism of its function based on the crystal structure of GAF domain (PDB 3C2W). The BV chromophore is highlighted in green. (**b**) Absorbance spectra for GAF-CaMP2 in Ca^2+^-bound and Ca^2+^-free state at pH 7.2. (**c**) Excitation and emission spectra for GAF-CaMP2 in Ca^2+^-bound and Ca^2+^-free states, pH 7.2. (**d**) Fluorescence intensity for GAF-CaMP2 in Ca^2+^-bound and Ca^2+^-free states as a function of pH. Three replicates were averaged for analysis. Error bars represent the standard deviation. (**e**) Ca^2+^ titration curves for GAF-CaMP2 in the absence and in the presence of 1 mM MgCl_2_, pH 7.2. Three replicates were averaged for analysis. Error bars represent the standard deviation. (**f**) Photobleaching curves for GAF-CaMP2 in Ca^2+^-bound state and smURFP. The power of light before the objective lens was 2.23 mW/cm^2^. Four replicates were averaged for analysis. (**g**) Maturation of purified GAF-CaMP2-sfGFP_sat_ fusion protein. The experimental data were fitted by the Hill equation. Protein maturation occurred in Ca^2+^-saturated buffer 30 mM HEPES, pH 7.20, 100 mM KCl, 1 µM BV, 5 mM Ca-EDTA at 37°C. The green (ex. 460/10 nm, em. 520/10 nm) and far-red (ex. 640/10 nm, em. 675/10 nm) fluorescence were simultaneously registered for sfGFP and GAF-CaMP2_sat_ in their fusion, respectively.

**Figure 4 ijms-20-03488-f004:**
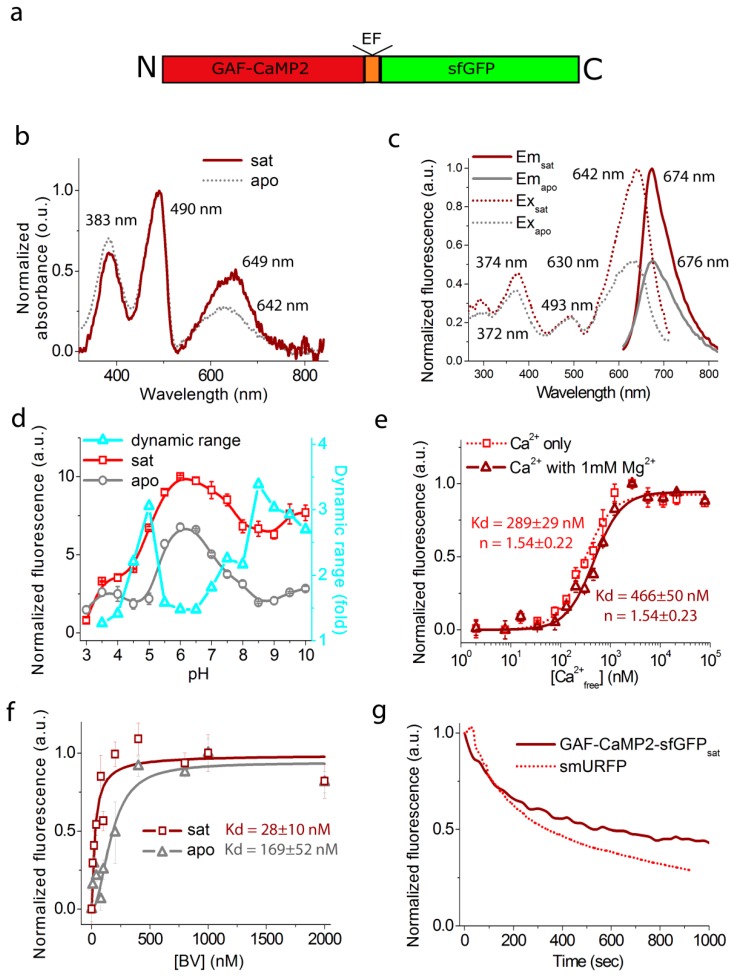
In vitro properties of the purified ratiometric GAF-CaMP2-sfGFP indicator. (**a**) A scheme for the fusion of GAF-CaMP2 indicator and sfGFP. (**b**) Absorbance spectra for GAF-CaMP2-sfGFP in Ca^2+^-bound and Ca^2+^-free states at pH 7.2. (**c**) Excitation and emission spectra for GAF-CaMP2-sfGFP in Ca^2+^-bound and Ca^2+^-free states, pH 7.2. (**d**) Fluorescence intensity for GAF-CaMP2 in its fusion with sfGFP in Ca^2+^-bound and Ca^2+^-free states as a function of pH. Three replicates were averaged for analysis. Error bars represent the standard deviation. (**e**) Ca^2+^ titration curves for GAF-CaMP2 in fusion with sfGFP in the absence and in the presence of 1 mM MgCl_2_, pH 7.2. Three replicates were averaged for analysis. Error bars represent the standard deviation. (**f**) BV titration of the GAF-CaMP2-sfGFP in Ca^2+^-bound and Ca^2+^-free states at pH 7.2. The experimental data were fitted by Hill equation. (**g**) Photobleaching curves for GAF-CaMP2 in its fusion with sfGFP in Ca^2+^-bound state and smURFP. The power of light before objective lens was 2.23 mW/cm^2^. Four replicates were averaged for analysis.

**Figure 5 ijms-20-03488-f005:**
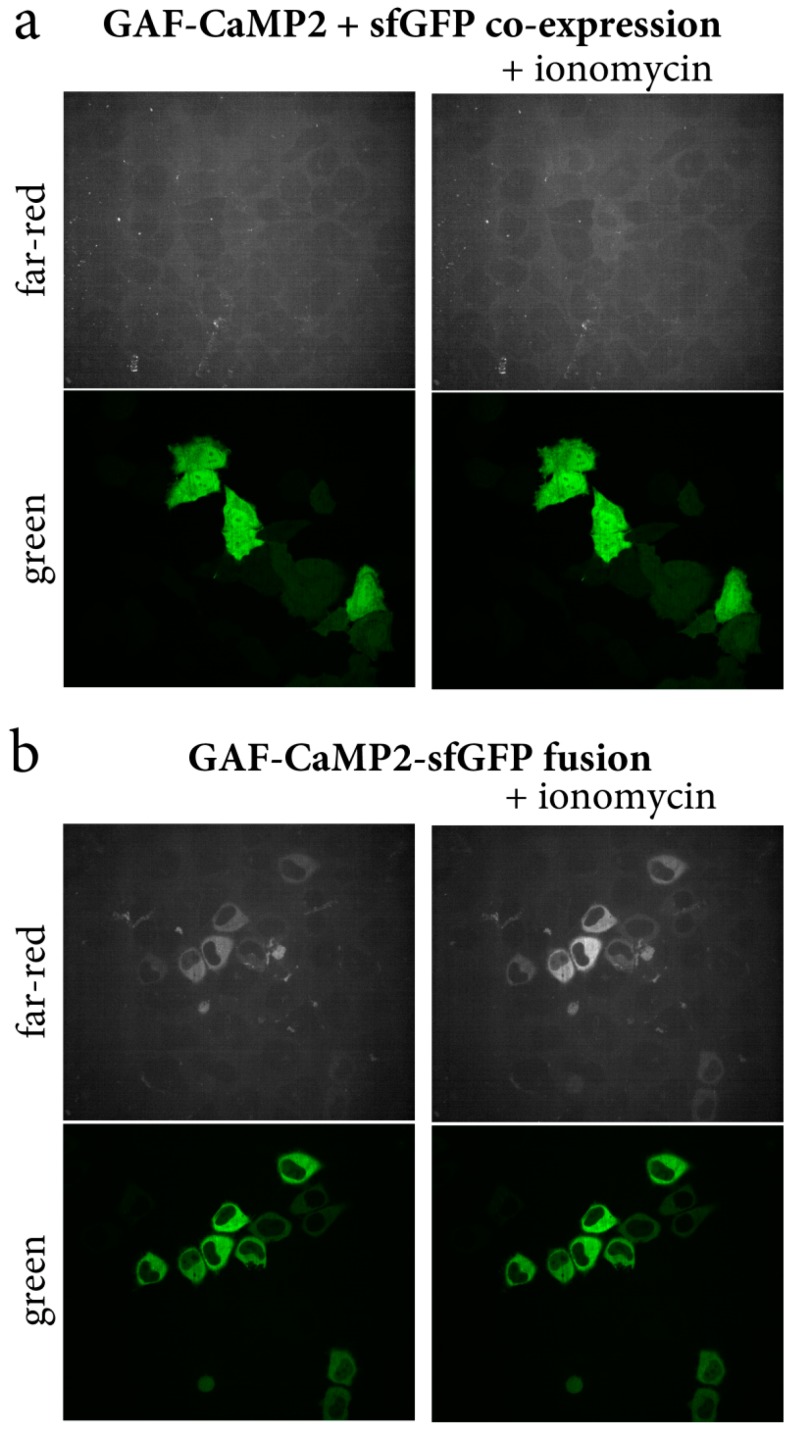
Optimization of GAF-CaMP2 expression in mammalian cells. Confocal images of HeLa Kyoto cells co-expressing green sfGFP protein and far-red NES-GAF-CaMP2 indicator (**a**) and expressing green/far-red NES-GAF-CaMP2-sfGFP fusion (**b**) before and after addition of the 2.5 µM ionomycin. A total of 20 µM BV was supplied during cell’s transfection procedure 24–48 h before imaging. Green (ex. 488 nm, em. 525/50 nm) and far-red (ex. 640 nm, em. 685/40 nm) channels are shown.

**Figure 6 ijms-20-03488-f006:**
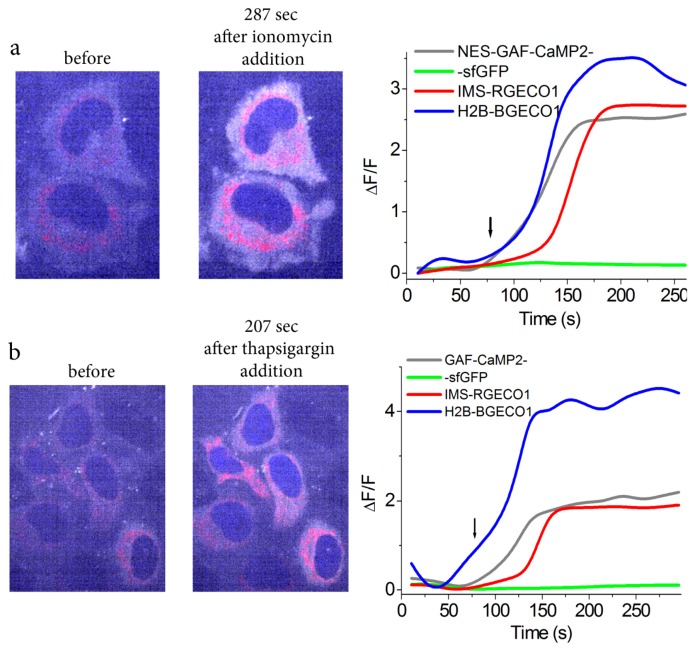
Visualization of calcium transients in three organelles of mammalian cells using GAF-CaMP2 indicator and four-color fluorescence confocal microscopy. Confocal images of HeLa Kyoto cells co-expressing blue H2B-B-GECO1, green/far-red NES-GAF-CaMP2-sfGFP, and red IMS-R-GECO1 calcium indicators in the nucleus, cytosol, and intermembrane space of mitochondria, respectively, before and respective time after addition of the 2.5 µM ionomycin (**a**) or 10 µM thapsigargin (**b**). Overlap images of blue (in blue, ex. 405 nm, em. 447/60 nm), red (in red, ex. 561 nm, em. 617/73 nm), and far-red (in grey, ex. 640 nm, em. 685/40 nm) channels are shown; the green (ex. 488 nm, em. 525/50 nm) channel is intentionally missed. The graphs illustrate ΔF/F changes in all fluorescence channels for respective indicators. Example of one cell is shown. The time of ionomycin and thapsigargin addition is shown by the arrow. A total of 20 µM BV was supplied during the cell’s transfection procedure 24–48 h before imaging.

**Table 1 ijms-20-03488-t001:** In vitro properties of the GAF-CaMP2 calcium indicator compared to permanently fluorescent smURFP and GAF-FP proteins.

Properties			Proteins					
			GAF-CaMP2		GAF-CaMP2-sfGFP		smURFP^a^	GAF-FP^b^
			apo	sat	apo	sat		
Absorbance maximum (nm)			642 (384)	650 (384)	630 (383, 490)	649 (383, 490)	642 *	637 (379)
Excitation maximum (nm)			620 (371)	640 (374)	630 (372, 493)	642 (374, 493)	642	635
Emission maximum (nm)			676	674	676 (514)	674 (514)	670	670
Quantum yield (%) ^c^			1.8 ± 0.2	4.8 ± 0.2	3.3 ± 0.1	6.9 ± 0.5	17.9 ± 0.2 *	7.3
ε (mM^−1^·cm^−1^) ^d^			22.2 ± 0.2	29.2 ± 0.3	15.7 ± 0.4	27.5 ± 0.9	180	49.8
Brightness vs. EGFP (%) ^e^			1.2	4.2	1.5	5.6	96	11
p*K*_a_			5.16 ± 0.05; 6.99 ± 0.06; ≥9.3 ± 0.3	4.89 ± 0.05; 7.15 ± 0.02; ≥9.1 ± 0.2	5.30 ± 0.02; 7.28 ± 0.02; ≥9.28 ± 0.06	4.89 ± 0.01; 7.60 ± 0.07; ≥9.31 ± 0.09	3.3 *	4.0; 7.8
ΔF/F (%)	Purified protein	0 mM Mg	103 ± 33		93 ± 11		NA	NA
		1 mM Mg	77 ± 16		78 ± 7			
	HeLa cells		non-fluorescent		169 ± 56			
K_d_ (nM) ^f^	0 mM Mg		235 ± 37 (0.88 ± 0.12)		289 ± 29 (1.54 ± 0.22)			
	1 mM Mg		435 ± 39 (1.48 ± 0.18)		466 ± 50 (1.54 ± 0.23)			
Maturation half-time (min)			ND		ND	28.2 ± 0.7 (sfGFP) 44 ± 2 (GAF-CaMP2)	39	240
Photobleaching half-time (s) ^g^			ND	1150 ± 560	ND	1080 ± 560	377 ± 55 *	ND

^a^ Data from [13]. Data marked with an asterisk were determined in this paper. NA, not applicable. ND, not determined. ^b^ Data from [12]. ^c^ QY for form with excitation maximum at 620−642 nm was determined at pH 7.20. miRFP720 was used as the reference standard [5]. ^d^ Extinction coefficient for form with absorbance maximum at 630−650 nm was determined relative to the Soret band at 384 nm for GAF-CaMP2 and relative to sfGFP form at 490 nm [29] for GAF-CaMP2-sfGFP. ^e^ Brightness was calculated as a product of the quantum yield and extinction coefficient and normalized to the brightness of EGFP that has an extinction coefficient of 56,000 M^−1^·cm^−1^ and quantum yield of 0.6 [30]. ^f^ Hill coefficient is shown in square brackets. In the absence and in the presence of 1 mM Mg^2+^ GCaMP6f had K_d_ values of 371 ± 20 nM (*n* = 2.4 ± 0.2) and 632 ± 15 nM (*n* = 2.25 ± 0.06), respectively. ^g^ The power of light from a mercury-xenon lamp before the objective lens was 2.23 mW/cm^2^.

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
