# Peer review of "Near-Infrared Genetically Encoded Positive Calcium Indicator Based on GAF-FP Bacterial Phytochrome"

_ijms, 2019, doi:10.3390/ijms20143488_

Round 1

Reviewer 1 Report

This an interesting and important paper describing a second near-IR genetically encoded Ca2+-sensor. 

There is one major issue that has to be fixed before publication. It seems that the quantum yields of GAF-CaMP2-sfGFP measured at two excitation wavelengths (630 nm and 384 nm) are very different (differ by 7 times). This is very unusual, because typically the excitation spectra of iRFPs are very similar to absorption spectra. The Soret peak is just a different electronic transition of the same chromophore. Therefore the authors either did not correct fluorescence spectra to the spectral sensitivity of the detector when measuring integrated fluorescence values or the quantum yields of mTagBF2 and/or miRFP720 that they use as standards are not correct. There are better, commonly used fluorescence standards for these regions of spectrum.

Minor issues: 

1) Figure 1 -  The authors might want to get rid of the residual red and blue wavy underlines.  

2) Figure 3b - change units of "o.u" to "a.u"

3) Page 11, line 9: The literature actually suggests that a lower affinity to Ca2+ would indicate faster kinetics. For example, the slow GCaMP6s has a Kd of 144 nM, while the fast GCaMP6f has a Kd of 375 nM (Chen et al. 2013, Nature). 

4) Page 11 Section 3 should be called Materials and Methods, not Discussion.

4) Do you have any possible explanation of why the fusion with sfGFP folds in mammalian cells, but the GAF-CaMP2 does not? 

Author Response

Response to Reviewer 1 Comments

We thank reviewer 1 for the valuable comments and useful suggestions, which we have addressed entirely in the revised manuscript. As requested by the reviewer 1 we added additional calculations, experiments and Figures to further characterize the novel GAF-CaMP2 calcium indicator. In particular, we carried out additional calculations and experiments to measure quantum yield of smURFP protein and compare the folding efficiency and in vitro stability for the smURFP protein and GAF-CaMP2 indicator alone and in fusion with sfGFP.

Overall, the revised manuscript has 3 new display items such as the Figures 1b, S8, and S9. In the main text we also expanded the Conclusion chapter.

Reviewer #1:

This an interesting and important paper describing a second near-IR genetically encoded Ca2+-sensor. 

Point 1: There is one major issue that has to be fixed before publication. It seems that the quantum yields of GAF-CaMP2-sfGFP measured at two excitation wavelengths (630 nm and 384 nm) are very different (differ by 7 times). This is very unusual, because typically the excitation spectra of iRFPs are very similar to absorption spectra. The Soret peak is just a different electronic transition of the same chromophore. Therefore the authors either did not correct fluorescence spectra to the spectral sensitivity of the detector when measuring integrated fluorescence values or the quantum yields of mTagBF2 and/or miRFP720 that they use as standards are not correct. There are better, commonly used fluorescence standards for these regions of spectrum.

Response 1: Although the excitation spectra of iRFPs are very similar to absorption spectra, usually the fluorescence occurs in appreciable yield only from the lowest excited state. This is in accordance with the Kasha’s rule (Kasha, M. (1950) Discussions of the Faraday Society). There are two major bands in the absorption spectra of iRFPs: the intense Soret band is attributed to the S0 to S2 transition whereas the Q bands are attributed to the S0 to S1 transition. According to the Kasha’s rule and Jablonski diagram (Jablonski, A. (1933) Nature) a fluorophore is usually excited to some higher vibrational level of either S1 or S2. After that usually molecules rapidly relax to the lowest vibrational level of S1, and fluorescence emission generally results from the thermally equilibrated lowest vibrational level of the S1 exited state. Usually the quantum yields for Soret band for the published biliverdin-based iRFP proteins are not reported. However, we found some examples for phytochrome derivatives, which characterize the quantum yield of Soret band relative to the quantum yield of Q band. For example, in the case of a phytochrome from oat shoots, the quantum yields for excitation at Soret and Q bands were practically equal (Colombano, C. G., et al. (1990) Photochemistry and Photobiology). In turn, for D207H mutant of DrBphP bacteriophytochrome, the quantum yield for Soret band was significantly higher as compared to quantum yield for Q band, as judged according to the absence of any fluorescence when excited at Q band (Wagner, J.R. et al. (2009) J. Biol. Chem.). For recently published NIR-GECO1 calcium indicator, two-photon imaging of cultured neurons expressing this protein showed neuronal-activity-dependent changes in NIR-GECO1 fluorescence when excited at 1250 nm characteristic for Q band, but when the authors used 880-nm excitation characteristic for Soret band, they did not observe fluorescence changes of NIR-GECO1 associated with Soret band (Yong Qian et al. (2019) Nature Methods). Therefore, according to the published data, fluorescent states corresponding to Soret and Q bands can behave differently, and can have either equal or different quantum yields.

            According to the manual for CM2203 spectrofluorometer (Solar, Belarus), which was used for the quantum yields measurements in this paper it determines true emission spectrum and makes automatic correction for quantum efficiency of the detector, relative width of monochromators bandwidth at constant width of slits and monochromator transmittance. This makes us confident that we measured corrected fluorescence spectra and calculated right integrated florescence values and quantum yields.

            The quantum yields for mTagBFP2 (Subach, O.M. et al. (2011) PloS one) and miRFP720 (Shcherbakova, D. M. et al. (2018) Nat. Chem. Biol.) proteins were measured relative to the quantum yields of Quinin sulfate and Nile blue dyes, which are commonly used fluorescence quantum yield standards. So we suggest that the published quantum yields for these proteins were measured correctly.

            Finally, we measured quantum yields for Soret and Q bands of smURFP protein using the same mTagBFP2 and miRFP720 proteins as quantum yields standards (Table 1). These values were practically the same of 0.18 and 0.179, respectively. The value of 0.179±0.02 coincided with published value of 0.18 for smURFP protein (Erik A. Rodriguez et al. (2016) Nature Methods). We think that these data evidence that we measured correct values for the quantum yield of GAF-CaMP2.

Minor issues: 

Point 2: 1) Figure 1 -  The authors might want to get rid of the residual red and blue wavy underlines.  

Response 2: In the revised manuscript, Figure 1a, we removed the residual red and blue wavy underlines.

Point 3: 2) Figure 3b - change units of "o.u" to "a.u"

Response 3: In the revised manuscript, Figure 3b, we corrected units from “o.u.” to “a.u.”.

Point 4: 3) Page 11, line 9: The literature actually suggests that a lower affinity to Ca2+ would indicate faster kinetics. For example, the slow GCaMP6s has a Kd of 144 nM, while the fast GCaMP6f has a Kd of 375 nM (Chen et al. 2013, Nature). 

Response 4: We agree that calcium indicators with lower calcium affinity have faster kinetics as compared to GECIs with higher calcium affinity. E.g., according to 10AP stimulation of neurons, GCaMP6f has 4.5-fold faster decay half-time (1796 ms) as compared to the same time for GCaMP6f (400 ms); GGaMP6f has 6-fold faster rise time (80 ms) vs rise time for GCaMP6s (480 ms) (Chen et al. (2013) Nature). These differences are seen on hundred-millisecond time scale. In our case, we detected difference on tens of seconds time scale (Figure 5), which cannot be attributed to the difference in calcium binding/dissociation kinetics for the B-GECO, GAF-CaMP2 and R-GECO1 indicators. Instead we suggest that R-GECO1 with lower affinity to calcium ions has lower sensitivity to calcium transients and weakly sense calcium transients at low calcium concentrations (50-100 nM in resting cell) and this is a reason why R-GECO1 starts to sense calcium increase with delay as compared to B-GECO and GAF-CaMP2 indicators. In other words, ionomycin- and thapsigargin-induced calcium transients in the cells are not so fast (on tens of seconds scale) and all indicators are capable to follow it, but R-GECO1 with lower calcium affinity because of lower sensitivity to calcium ions starts to maximally change its fluorescence at higher calcium concentrations.

            Accordingly, in the revised manuscript Page 11, we added “ …or sensitivity …”

Point 5: 4) Page 11 Section 3 should be called Materials and Methods, not Discussion.

Response 5: In the revised manuscript, we renamed section 3 and now it is called as Materials and Methods.

Point 6: 4) Do you have any possible explanation of why the fusion with sfGFP folds in mammalian cells, but the GAF-CaMP2 does not? 

Response 6: In the revised manuscript, Conclusions section, we added: “The fusion of GAF-CaMP2 indicator with sfGFP folded in mammalian cells, but the GAF-CaMP2 alone does not. We suggest several possible reasons for the extremely weak fluorescence of GAF-CaMP2 indicator in mammalian cells related to its possible poor folding, low stability or susceptibility to degradation by cellular machinery. It is known that sfGFP protein or other well-folded proteins may facilitate folding of proteins in fusion with them [37]. In accordance with this assumption the extinction coefficient for bacterially expressed GAF-CaMP2 protein (3987 M-1 cm-1 at 650 nm) calculated relative to the absorption peak at 280 nm was 2.6-fold smaller as compared to the extinction coefficient for GAF-CaMP2-sfGFP fusion (10192 M-1 cm-1 at 649 nm). Hence, in bacterial cells GAF-CaMP2 protein folds 2.6-fold less efficiently as compared to its fusion with sfGFP. Therefore, GAF-CaMP2 alone may fold inefficiently in mammalian cells in similar way as in bacteria and sfGFP fused to its C-terminal end may improve its folding by 2.6 times. High cellular stability of bacteriophytochrome-based proteins can also correlate with their high protein stability in vitro [39]. Respectively, the possible poor stability of GAF-CaMP2 protein may also affect its brightness in mammalian cells. We compared the in vitro stability of GAF-CaMP2, GAF-CaMP2-sfGFP and smURFP proteins according to resistance of their fluorescence to guanidine hydrochloride denaturating reagent. We found that both GAF-CaMP2 and GAF-CaMP2-sfGFP had higher in vitro stability as compared to smURFP protein ((ESI Figure S9). GAF-CaMP2 demonstrated in vitro stability in fusion with sfGFP, similar to its stability alone (ESI Figure S9). Hence, GAF-CaMP2 itself has high stability in vitro and sfGFP protein practically does not affect it and can not increase expression level of fused GAF-CaMP2 protein in mammalian cells by increasing of GAF-CaMP2 stability. Some peptides, called degrons, in fusion with protein of interest (POI) can modulate the intracellular POI concentration through the susceptibility to degradation via cellular proteasome system [40]. sfGFP may serve as stabilizing degron tag and elongate half-life of the GAF-CaMP2 protein in mammalian cells. Hence, sfGFP enhances expression level of fluorescent GAF-CaMP2 indicator in mammalian cells, probably, by improving folding of GAF-CaMP2 and/or the decreasing its susceptibility to degradation.”.

Reviewer 2 Report

This paper introduces the design of a genetically-encoded calcium sensor that exhibits fluorescence in the near-infrared region and is an alternative to the recently developed NIR-GECO1 by the Campbell group. The work has been carefully conducted, is well presented, and the data support the conclusions drawn. A nice application using four-color fluorescence microscopy is presented. Although the presented GAF-CaMP2 has significant drawbacks (notably a rather low dynamic range) which may limit its outreach in its current version, it represents a novel successful attempt to develop a fluorescent calcium sensor based on bacterial-cytochromes-derived near-infrared fluorescent proteins. The non-inverted response to calcium transients, which limits photobleaching issues, and the fluorescence excitation peaking at 642 nm, close to a common laser band, are clear advantages relative to NIR-GECO1. Provided the following questions are properly addressed, I recommend publication in IJMS.

1/ One puzzling question concerns the mechanism behind the “activation” property of sfGFP in mammalian cells when this FP is fused to GAF-CaMP2. The authors do not give any hint of what may be going on, and it is rather disappointing not to see any discussion at the end of the paper on this rather puzzling matter.

2/ it would also be nice to comment on the mechanism behind positive versus inverted response to calcium ions. Please add a comparison of NIR-GECO1 with GAF-CaMP2 and try to address the structural reasons for such an opposite response in the two sensors.

P2/L16: check sentence

P2/L6: at “most” commercial microscopes (rather than “all”)

P2/L14-17: why do insertion attempts need to be positioned close to aromatic residues ? Please provide a hint for this.

Figure 1: I would suggest that to add a structural view (of PaBphP) showing the different insertion sites tested.

P6/L6 => call to figure 3, not figure 1 ?

P12/Discussion: in fact this paragraph is not Discussion but Materials and Methods. So in the end there is essentially no discussion, which is regrettable. Please consider to extend the Conclusion of the paper, notably to add comments relative to points 1 and 2 above.

FigS3: the error bars in panel C are very difficult to see. Please improve this figure.

The supplementary results and discussion section could be incorporated into the main manuscript. I don't see the interest of this small paragraph being shifted away in the SI.

Author Response

Response to Reviewer 2 Comments

We thank reviewer 2 for the valuable comments and useful suggestions, which we have addressed entirely in the revised manuscript. As requested by the reviewer 2 we added additional calculations, experiment and Figures to further characterize the novel GAF-CaMP2 calcium indicator. In particular, we carried out additional calculations and experiment to compare the folding efficiency and in vitro stability for smURFP protein and the GAF-CaMP2 indicator alone and in fusion with sfGFP.

Overall, the revised manuscript has 3 new display items such as the Figures 1b, S8, and S9. In the main text we also expanded the Conclusion chapter.

Reviewer #2:

This paper introduces the design of a genetically-encoded calcium sensor that exhibits fluorescence in the near-infrared region and is an alternative to the recently developed NIR-GECO1 by the Campbell group. The work has been carefully conducted, is well presented, and the data support the conclusions drawn. A nice application using four-color fluorescence microscopy is presented. Although the presented GAF-CaMP2 has significant drawbacks (notably a rather low dynamic range) which may limit its outreach in its current version, it represents a novel successful attempt to develop a fluorescent calcium sensor based on bacterial-cytochromes-derived near-infrared fluorescent proteins. The non-inverted response to calcium transients, which limits photobleaching issues, and the fluorescence excitation peaking at 642 nm, close to a common laser band, are clear advantages relative to NIR-GECO1. Provided the following questions are properly addressed, I recommend publication in IJMS.

Point 1: 1/ One puzzling question concerns the mechanism behind the “activation” property of sfGFP in mammalian cells when this FP is fused to GAF-CaMP2. The authors do not give any hint of what may be going on, and it is rather disappointing not to see any discussion at the end of the paper on this rather puzzling matter.

Response 1: In the revised manuscript, we combined Discussion with Results and “Results” section is now called as “Results and Discussion”.

            In the revised manuscript, Conclusions section, we added: “The fusion of GAF-CaMP2 indicator with sfGFP folded in mammalian cells, but the GAF-CaMP2 alone does not. We suggest several possible reasons for the extremely weak fluorescence of GAF-CaMP2 indicator in mammalian cells related to its possible poor folding, low stability or susceptibility to degradation by cellular machinery. It is known that sfGFP protein or other well-folded proteins may facilitate folding of proteins in fusion with them [37]. In accordance with this assumption the extinction coefficient for bacterially expressed GAF-CaMP2 protein (3987 M-1 cm-1 at 650 nm) calculated relative to the absorption peak at 280 nm was 2.6-fold smaller as compared to the extinction coefficient for GAF-CaMP2-sfGFP fusion (10192 M-1 cm-1 at 649 nm). Hence, in bacterial cells GAF-CaMP2 protein folds 2.6-fold less efficiently as compared to its fusion with sfGFP. Therefore, GAF-CaMP2 alone may fold inefficiently in mammalian cells in similar way as in bacteria and sfGFP fused to its C-terminal end may improve its folding by 2.6 times. High cellular stability of bacteriophytochrome-based proteins can also correlate with their high protein stability in vitro [39]. Respectively, the possible poor stability of GAF-CaMP2 protein may also affect its brightness in mammalian cells. We compared the in vitro stability of GAF-CaMP2, GAF-CaMP2-sfGFP and smURFP proteins according to resistance of their fluorescence to guanidine hydrochloride denaturating reagent. We found that both GAF-CaMP2 and GAF-CaMP2-sfGFP had higher in vitro stability as compared to smURFP protein ((ESI Figure S9). GAF-CaMP2 demonstrated in vitro stability in fusion with sfGFP, similar to its stability alone (ESI Figure S9). Hence, GAF-CaMP2 itself has high stability in vitro and sfGFP protein practically does not affect it and can not increase expression level of fused GAF-CaMP2 protein in mammalian cells by increasing of GAF-CaMP2 stability. Some peptides, called degrons, in fusion with protein of interest (POI) can modulate the intracellular POI concentration through the susceptibility to degradation via cellular proteasome system [40]. sfGFP may serve as stabilizing degron tag and elongate half-life of the GAF-CaMP2 protein in mammalian cells. Hence, sfGFP enhances expression level of fluorescent GAF-CaMP2 indicator in mammalian cells, probably, by improving folding of GAF-CaMP2 and/or the decreasing its susceptibility to degradation.”.

Point 2: 2/ it would also be nice to comment on the mechanism behind positive versus inverted response to calcium ions. Please add a comparison of NIR-GECO1 with GAF-CaMP2 and try to address the structural reasons for such an opposite response in the two sensors.

Response 2: In the revised manuscript, Conclusion section, we added: “GAF-CaMP2 and NIR-GECO1 have positive and inverted responses to calcium ions, respectively. To suggest possible structural bases for such an opposite response in two sensors, we aligned the X-ray structures of the PaBphP and DrBphP bacteriophytochromes and found that the insertion sites of calcium-binding domains for GAF-CaMP2 and NIR-GECO1 indicators are located on opposite sides of the BV chromophore (ESI, Figures S8). The opposite arrangement of calcium-binding domains relative to the BV chromophore for GAF-CaMP2 and NIR-GECO1 may be one of the possible explanations for the opposite calcium response for these indicators.”

Point 3: P2/L16: check sentence

Response 3: In the revised manuscript we replaced “A possible reason for the limited number of NIR calcium indicators could be that …” with “A possible reason for the limited number of NIR calcium indicators may be due to the fact that …”

Point 4: P2/L6: at “most” commercial microscopes (rather than “all”)

Response 4: In the revised manuscript, we replaced “…all commercial microscopes.” with “…most commercial microscopes.”.

Point 5: P2/L14-17: why do insertion attempts need to be positioned close to aromatic residues ? Please provide a hint for this.

Response 5: In the revised manuscript, we added “; we suggested that changing the position of the bulky aromatic residue attached to the calcium-binding domain through the linker may strongly influence the fluorescence of the BV chromophore.”

Point 6: Figure 1: I would suggest that to add a structural view (of PaBphP) showing the different insertion sites tested.

Response 6:  In the revised manuscript, Figure 1, we added panel b showing BaBphP GAF-domain structure with insertion sites which were probed while developing the GAF-CaMP calcium indicator.  

Point 7: P6/L6 => call to figure 3, not figure 1 ?

Response 7: In the revised manuscript, we now refer to figure 3 not figure 1.

Point 8: P12/Discussion: in fact this paragraph is not Discussion but Materials and Methods. So in the end there is essentially no discussion, which is regrettable. Please consider to extend the Conclusion of the paper, notably to add comments relative to points 1 and 2 above.

Response 8: In the revised manuscript, we renamed section 3 and now it is called as Materials and Methods.

            In the revised manuscript, we extended the Conclusions section of the paper by adding the comments to points 1 and 2 above.

Point 9: FigS3: the error bars in panel C are very difficult to see. Please improve this figure.

Response 9: In the revised manuscript, we changed color of the error bars in panel c and now they are visible.

Point 10: The supplementary results and discussion section could be incorporated into the main manuscript. I don't see the interest of this small paragraph being shifted away in the SI.

Response 10: In order not to mislead the reader, we want to keep this results and discussion section in supplementary section because they describe the first version of GAF-CaMP not final version called GAF-CaMP2.

Round 2

Reviewer 1 Report

The authors did not reply to my main question: Why does the new construct violate the Kasha's rule? 

To answer this question, they should provide the absorption and excitation spectra of the GAF-CaMP2-sfGFP construct, similarly to what they do in Fig. 3, b,c for the GAF-CaMP2 protein. The excitation spectrum should be corrected for the excitation intensity dependence on wavelength in this measurement. The ratio of the absorption peaks (Q vs Soret) should be compared to the ratio of corresponding excitation peaks. If they are different by a factor of ~7, then this can explain the discrepancy in the effective quantum yields. (This would mean that the majority of the 385 nm species are not fluorescent). In this case the value that they measure upon 385-nm excitation cannot be called quantum yield. If excitation and absorption spectra have similar shape, than there should be another reason. This could be, for example, a method of how they correct for the equal excitation bandwidths of the fluorimeter at different frequencies (is this correction really needed?) or the absence of correction for PMT spectral sensitivity. In that latter case, the signal of mTagBFP would be overestimated compared to that of miRFP720 because of the sharp decrease of some PMT's sensitivity towards infrared and this would underestimate the quantum yield for 385-nm excitation.

These possible explanations should be considered before the publication. 

Other, minor questions were addressed appropriately.

Author Response

Response to Reviewer 1 Comments

We thank reviewer 1 for the raised point, which we tried to address.

Reviewer #1:

Point 1: The authors did not reply to my main question: Why does the new construct violate the Kasha's rule? 

To answer this question, they should provide the absorption and excitation spectra of the GAF-CaMP2-sfGFP construct, similarly to what they do in Fig. 3, b,c for the GAF-CaMP2 protein. The excitation spectrum should be corrected for the excitation intensity dependence on wavelength in this measurement. The ratio of the absorption peaks (Q vs Soret) should be compared to the ratio of corresponding excitation peaks. If they are different by a factor of ~7, then this can explain the discrepancy in the effective quantum yields. (This would mean that the majority of the 385 nm species are not fluorescent). In this case the value that they measure upon 385-nm excitation cannot be called quantum yield. If excitation and absorption spectra have similar shape, than there should be another reason. This could be, for example, a method of how they correct for the equal excitation bandwidths of the fluorimeter at different frequencies (is this correction really needed?) or the absence of correction for PMT spectral sensitivity. In that latter case, the signal of mTagBFP would be overestimated compared to that of miRFP720 because of the sharp decrease of some PMT's sensitivity towards infrared and this would underestimate the quantum yield for 385-nm excitation.

These possible explanations should be considered before the publication. 

Other, minor questions were addressed appropriately.

Response 1: The new construct does not violate the Kasha’s rule because in accordance with the Kaha’s rule the fluorescence of GAF-CaMP2 indicator mainly occurs from S0-S1 transition (quantum yield of 6.8% for Q-band in sat-state) and with lower efficiency from S0-S2 transition (quantum yield of 1% for Soret band in sat-state).

            In Supplementary Information, Figure S4, the suggested by reviewer 1 absorption and excitation spectra for GAF-CaMP2-sfGFP protein are presented. The ratio of the absorption peaks (Q to Soret) and respective excitation peaks are 0.772 and 2.22, respectively. So, these ratios are different by a factor of 2.9.

All corrections for PMT sensitivity and others were made by the manufacturer of CM2203 spectrofluorometer (Solar, Minsk, Belarus). So we can not judge for them.

If our response is not adequately addressed the major concern of reviewer 1 we are ready to remove the estimation of the quantum yield values for the Soret band from the Table 1 because we do not mention and discuss it anywhere in the text.

Round 3

Reviewer 1 Report

Since the GAF-CaMP2-sfGFP is most interesting protein in terms of its possible applications in mammalian cells, it would make more sense to present Fig S4 in the main text. 

When calculating quantum yield of the GAF-CaMP2-sfGFP upon excitation at 590 nm, the fluorescence should be integrated in a wider range not between 624 and 820 nm, because the fluorescence intensity is clearly non-zero at wavelengths shorter than 624 and longer than 820 nm, so not the whole band is integrated. 

If the authors would like to evaluate the quantum yield upon excitation at the Soret band, the whole fluorescence spectrum (400 - 850 nm) should be presented in Fig. S4 for this excitation wavelength. Next, the authors should specify what particular specie they considering, sfGFP or the part containing BV chromophore. This would define the fluorescence integration region - either in the green or in the red/infrared parts, respectively. If the BV quantum yield is considered, the authors should make sure that at the excitation wavelength absorption of sFGFP is negligible (including abosrption of the neutral form). 

Author Response

Response to Reviewer 1 Comments

Reviewer #1:

Point 1: Since the GAF-CaMP2-sfGFP is most interesting protein in terms of its possible applications in mammalian cells, it would make more sense to present Fig S4 in the main text. 

Response 1: In the revised manuscript we presented Figure S4 in the main text.

Point 2: When calculating quantum yield of the GAF-CaMP2-sfGFP upon excitation at 590 nm, the fluorescence should be integrated in a wider range not between 624 and 820 nm, because the fluorescence intensity is clearly non-zero at wavelengths shorter than 624 and longer than 820 nm, so not the whole band is integrated. 

Response 2: In the revised manuscript we recalculated the quantum yield for the GAF-CaMP2-sfGFPapo and GAF-CaMP2-sfGFPsat upon excitation at 590 nm using wider range of 600 – 820 nm. The new values were practically the same as compared to the estimation of the quantum yield for the 624-820 nm range, i.e. 3.3±0.1 and 6.9±0.5 vs 3.2±0.2 and 6.8±0.1, respectively. We could not extend the 820 nm border because of limitations of the spectrofluorometer.

Point 3: If the authors would like to evaluate the quantum yield upon excitation at the Soret band, the whole fluorescence spectrum (400 - 850 nm) should be presented in Fig. S4 for this excitation wavelength. Next, the authors should specify what particular specie they considering, sfGFP or the part containing BV chromophore. This would define the fluorescence integration region - either in the green or in the red/infrared parts, respectively. If the BV quantum yield is considered, the authors should make sure that at the excitation wavelength absorption of sFGFP is negligible (including abosrption of the neutral form). 

Response 3: In the revised manuscript we did not evaluate the quantum yield for the Soret band.

Round 4

Reviewer 1 Report

All the issues were addressed by the authors and the manuscript can be published as is.